# Social and clinical risk factors associated with hospitalized COVID-19 patients in Brussels's deprived and multiethnic areas

Judith Racape[1,2]*, Anne-Cecile Noel[3], Jerome Lurel[4], Nicolas Dauby[5,6], Yves Coppieters[1], Jean-Christophe Goffard[4], Andrea Rea[7]

**1** Research Center in Epidemiology, Biostatistics and Clinical Research, School of Public Health, Universite libre de Bruxelles (ULB), Brussels, Belgium, **2** Chair in Health and Precarity, Faculty of Medecine, Universite libre de Bruxelles (ULB), Brussels, Belgium, **3** Social Department, Centre Hospitalier Universitaire Saint-Pierre, Université Libre de Bruxelles (ULB), Brussels, Belgium, **4** Department of Internal Medicine, Erasmus Hospital, Université Libre de Bruxelles (ULB), Brussels, Belgium, **5** Environmental Health Research Centre Public Health School, Université libre de Bruxelles (ULB), Brussels, Belgium, **6** Department of Infectious Diseases, Centre Hospitalier Universitaire (CHU) Saint-Pierre, Université Libre de Bruxelles (ULB), Brussels, Belgium, **7** Group for Research on Ethnic Relations, Migrations & Equality, Faculty of Philosophy and Social Sciences, Universite libre de Bruxelles (ULB), Brussels, Belgium

* judith.racape@ulb.be

**Data Availability Statement:** All the data have been included in the paper.

## Abstract

### Background

In Belgium, the Brussels-Capital region was severely affected by the COVID-19 epidemic. Various hypotheses were mentioned in order to explain Brussels' excess disease spreading and mortality rate, but socioeconomic risk factors are increasingly recognized. This study's objective was to analyze clinical and social profiles of patients hospitalized for COVID-19, by nationality groups, in two hospitals located in Brussels's deprived and multiethnic areas.

### Methods

Data covered hospitalized COVID-19 patients from two Brussels hospitals (n = 787) between the 1st of March 2020 and the 31st of June 2020. Social data was collected using hospital records, and clinical data was extracted from hospitals' COVID-19 databases. Multivariable logistic regression models were used to estimate the odds ratios (OR) of the association between two outcomes (Intensive Care Unit admission and mortality) and risk factors (social and clinical).

### Results

Patients from Sub-Saharan Africa were younger, had a higher prevalence of obesity, lacked health insurance, and had the highest proportion of Intensive Care Unit (ICU) admission (27.7%) but the lowest mortality rates than other nationality groups. Patients from North Africa had a higher prevalence of diabetes compared to other nationality groups and a high proportion of European patients came from nursing homes. Patients deprived of health insurance had a higher risk of ICU admission compared to those who had insurance (OR

**Funding:** The authors received no specific funding for this work.

**Competing interests:** The authors have declared that no competing interests exist.

IC95%; 1,9 1.1–3.6, p = 0.03). Other risk factors as sex and obesity were significantly associated to ICU admission and, age and hypertension were significantly associated to mortality.

## Conclusion

Social and clinical profile of the patients differs between the nationality groups, and some risk factors for Intensive Care Unit admission and mortality were linked to more patients' precarious situation as the availability of health insurance. This study underlines the role of selected social health determinants and the importance of routinely collecting social along with clinical data.

## Introduction

Belgium was one of the countries worst affected by the COVID-19 epidemic in Europe during the first wave with a monthly excess mortality rate of more than 50%, from late February to late June 2020 [1]. The Brussels-Capital region was particularly impacted, with an excess mortality rate of 81.7% (and 123% during the first wave), more than twice as high as the other two regions of Belgium [2]. In 2020, 649,223 confirmed cases of SARS-CoV-2 infection were reported in Belgium, including 81,655 in Brussels, i.e. nearly 13% [3]. Despite its relatively small size (1 218 255 inhabitants the 1rst January 2020), the Brussels-Capital region presented excess mortality during the first wave of COVID-19 in Brussels that was comparable to that of several larger European cities such as Paris or London. Various hypotheses were mentioned in order to explain Brussels' excess disease spreading and mortality rate: high population density, a very high percentage of people with immigration background (74,1% in 2019, compared to 31,1% on average in Belgium), with or without Belgian citizenship, travelling in and out of Brussels, due to its role as European capital [4]. Another specificity of Brussels-Capital is its high proportion of elderly people in care homes, compared to other regions [2]. Those establishments were severely hit by COVID-19 [3]. Brussels is also a multicultural city where one in three people live below the poverty line: the risk of poverty is therefore high, and access to care is also less than optimal. A large part of Brussels' population lives in precarious socio-economic conditions, which are linked to poor health [3]. All those difficult conditions possibly contribute to the disease spreading and worsened its impact on vulnerable people. It is now clearly established that older age, male sex, and a range of co-morbidities such as hypertension, cardiovascular disease, obesity, and diabetes are risk factors for severe or fatal COVID-19 [5–7]. However, studies increasingly insist on socio-economic risk factors [8–10]. In general, regions that have experienced very strong increases in mortality have a high poverty rate [11–13]. Seine-St-Denis, the most deprived department in France, experienced an excess mortality rate of 134% between March and April 2020 [11]. New York City suffered the greatest burden of deaths in the US during the first wave [14]. Previous studies also reported high rates of COVID-19 hospitalizations and mortality among racialized/ethnic minorities [15–17], crowded households [8, 18] and homeless shelters [19, 20]. Evidence suggests that ethnic minorities in urban settings tend to live in more crowded conditions but are also more likely to be employed in public-facing occupations, increasing the risk of virus transmission [14, 21]. COVID-19 has been characterized as a syndemic pandemic referring to the interaction between diseases, biological, social, and environmental factors that, when combined, worsen the impact of the disease on a specific population [22, 23].

In a Belgium study, a significant association between COVID-19 incidence and area deprivation was found, with the incidence in the most deprived areas predicted to be 24% higher than in the least deprived areas [24]. Another study in Belgium, focusing on income inequalities, have shown higher excess mortality from COVID-19 with lower income groups [25]. Concerning, ethnic minority communities, at our knowledge, only one study in Belgium shows excess mortality for middle-aged Belgians and Sub-Saharan African men. Whilst most male elderly migrant groups showed higher mortality than non-migrants, as opposed to 2019 and to women [26].

Limited research has been conducted on more vulnerable populations, particularly those without legal status and who are excluded from official statistics. And the absence of social data in clinical databases did not allow us to have hospitalized patients' overall clinical and social profiles of this population mostly "invisible". Our study therefore focused on two Brussels hospitals, CHU St Pierre and Erasme, located the first in the most deprived area of Brussels and the second in a deprived area. The least economically advantaged populations have been staying in this same area for decades, in North-West Brussels, one of the poorest areas in the region. In Brussels, the municipalities with the highest reported cases of SARS-CoV-2 in proportion to the population are located in the most deprived and densely populated areas [3]. The CHU Saint-Pierre is a also a public hospital to which most immigrants or uninsured people residing in the Brussels area are referred for treatment. This study's aim was to analyze the profile of patients hospitalized for COVID-19 during the first wave in both hospitals, which looked after a particularly precarious foreign origin population. More specifically, the objectives were 1/ to describe social and clinical profiles of the hospitalized population by nationality groups 2/ to analyze clinical and social risk factors associated with ICU and mortality taking into account socio-demographic features not routinely collected during the first wave of the national surveillance system [27].

## Methods

### Population

Data concerned all hospitalized COVID-19 patients from two Brussels hospitals (n = 786) between the 1st of March 2020 and the 31st of June, 2020. The Centre Hospitalier Universitaire Saint-Pierre (CHU Saint-Pierre) in Brussels is a tertiary reference hospital for infectious diseases and was the reference center for COVID-19 at the beginning of the pandemic (n = 318). Erasme hospital is the academic medical center of Université libre de Bruxelles (ULB) (n = 468).

All admitted individuals (age ≥ 18 years old) with either a confirmed positive SARS-CoV-2 polymerase chain reaction or high clinical suspicion for COVID-19, based on clinical presentation and computed tomography imaging of the chest (n = 786). We have excluded pregnant women (n = 36) and patients registered but not hospitalized (n = 83) (Fig 1).

### Data collection

Clinical data was retrospectively collected by the hospitals from their own COVID-19 database. Data from Erasme Hospital of Brussels used the International Severe Acute Respiratory and Emerging Infections Consortium (ISARIC) COVID-19 database [28]. Social and demographic data was collected on hospital record and linked through a unique encrypted number to extracted clinical data. Data from CHU Saint Pierre were collected in accordance with the Belgian national surveillance system [27]. Data were collected and managed using REDCap (Research Electronic Data Capture), a secure, web-based software platform that allows data collection for research studies [29].

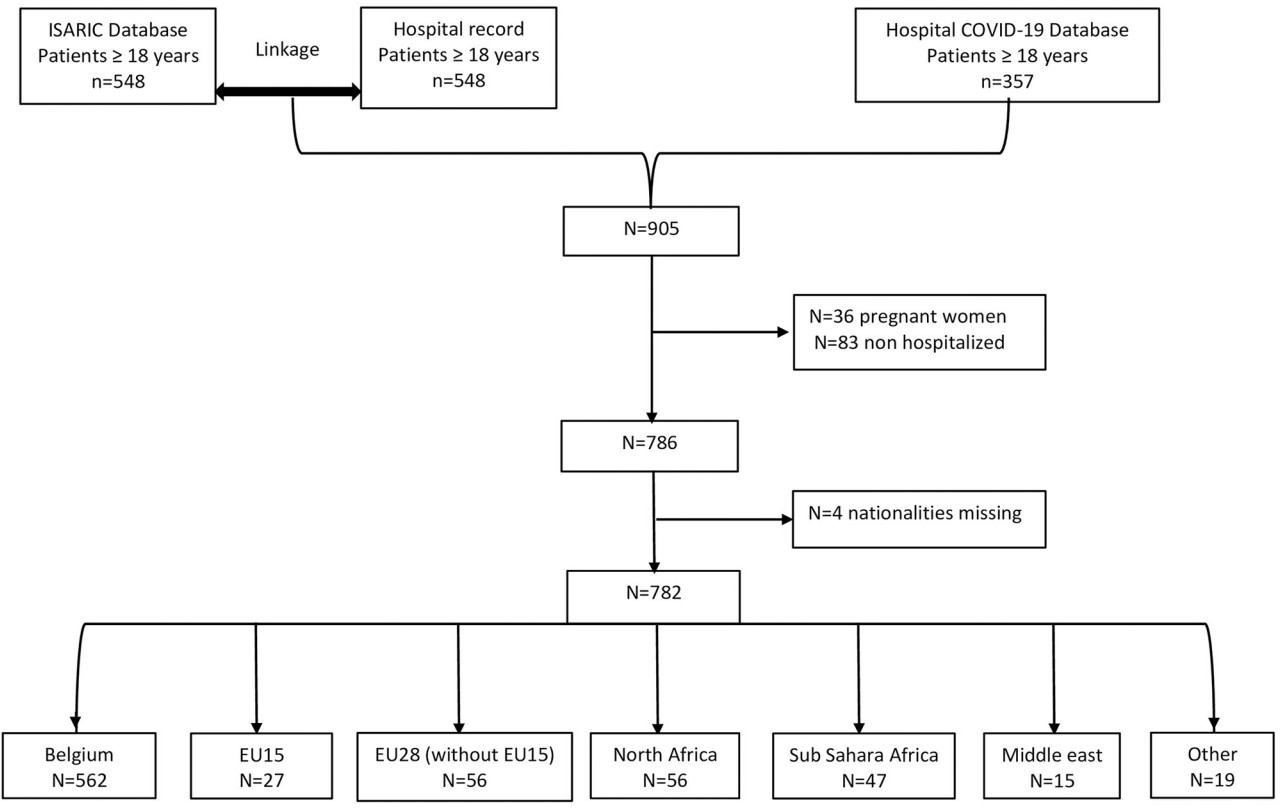

**Fig 1. Flow diagram of study participants.**

## Variables

Social and demographic data was collected on hospital records: sex, age, health insurance, postcode, nationality, origin of patients before admission.

Age was as under 65 and over 65 years old. In Belgium, most people retire at 65 (legal retirement age).

We used nationality at the hospital admission. Nationalities were classified in 7 groups, representing the most frequent nationalities in Brussels: Belgium, EU15, EU28 (excluding EU15), North Africa, Sub-Saharan Africa, Middle East, other. We separated EU28 in two groups, EU15 and the new member of EU28. This separation was made because the latter group includes a significant new population in Brussels (in 2020, Romania is the second most prevalent nationality in Brussels, following the French). This group comprises a high proportion of people working in essential occupations as cleaning caregiving, and construction [30] We did not interpret the category "other" because of its heterogeneity.

Belgium is divided into three regions:: Wallonia, Flanders and Brussels-Capital. Postcodes were divided in 4 groups: Flanders region, Wallonia region, Brussels-Capital Region' deprived area and Brussels Capital region' other areas. We pooled together the municipalities within Brussels-Capital Region' that are part of the "poor crescent", the most deprived area in Brussels [3].

Health insurance was categorized in 2 groups, with or without health insurance. "With insurance" category included people affiliated with a health care insurer. In the "without insurance" category, we included people without any health insurance and with temporary health

insurance (Urgent Medical Card, Federal agency for the reception of asylum seekers), given to transit migrants and asylum seekers, meaning people without or temporary residence permit. People with Urgent Medical care can only access healthcare through a complex and sometimes long administrative procedure, involving a social enquiry to verify their "illegal" stay and destitution. The complexity of the procedure often act as a barrier to accessing healthcare and tests equivalent to those without health insurance [31].

Origins of patients before admission was categorized as such: home, nursing home, shelter, other hospital units and other hospitals.

We extracted the clinical data: co-morbidities (obesity, hypertension, diabetes, cardiovascular disease, pulmonary disease and neoplasia), baseline laboratory parameters (C-reactive protein (CRP), partial pressure of oxygen (paO2), Lactate dehydrogenase (LDH), Lymphocytes), and clinical features (length of stay, admission in Insensitive Care Unit and mortality). Baseline laboratory parameters of disease severity consisting of LDH $\geq$ 350 IU/L, CRP $\geq$ 150 mg/L Lymphocytes < 500 and paO2 < 60 mmHg. Due to the high proportion of missing value for LDH and PaO2 (51% and 42% respectively), the interpretation of the results was limited. Therefore, we separated the results into a S1 Table.

## Statistical analysis

Qualitative variables were presented with proportions, and quantitative variables, such as length of stay, were presented with medians (P25-P75). Qualitative variables were compared by using chi square test or exact test. Non parametric quantitative variable (length of stay) was compared by using Kruskal Wallis test. The analysis covered two outcomes: admission in Insensitive Care Unit (ICU) and mortality. A logistic regression was used to estimate the odds ratios (ORs) of the association between outcomes and risk factors (social and clinical). Univariate and multivariable logistic models were constructed. The multivariable model included age, sex, health insurance, nationality, region and co-morbidities (hypertension, diabetes, cardiovascular disease, pulmonary disease and obesity). We did not include data prior to admission because patients from other hospital units and other hospitals were transferred to Erasme/ CHU St Pierre hospital due to their critical state. Those variables are an intermediate factor for ICU admission and mortality.

Crude and adjusted Odd Ratios with 95% Confidence Interval derived from the logistic regressions and the *p*-value of the Wald chi$^2$ test were presented. The Hosmer et Lemeshow test was used to check models' adjustment. The significance level was set at $\alpha$ = 0.05. Analyses were performed using the Stata software, version16.

## Ethical approval

The CHU Saint-Pierre and Erasme hospital's Ethical Committee approved this study (CE/21-01-03 and P2021/178). A statement that formal consent was not obtained because the extracted data was anonymous and we linked the data through a unique encrypted number.

## Results

### Characteristics of people hospitalized for COVID-19 by nationality groups

Table 1 shows the characteristics of people hospitalized for COVID-19 by nationality groups. We observe significant differences in terms of age, health insurance, region, patient origins, hypertension, cardiovascular disease and pulmonary disease between migrant groups. Patients from sub-Saharan Africa were younger (78.7% under 65 years) with a higher prevalence of obesity (42.6%), and lacked health insurance (53.2%) as compared to other nationality groups.

**Table 1. Characteristics of the population hospitalized for COVID-19 by nationality groups.**

| n (%) | Total N = 782 | Belgium n = 562 | EU15 n = 27 | EU28 (without EU15) n = 56 | North Africa N = 56 | Sub-Sahara Africa N = 47 | Middle east N = 15 | p-value |
|---|---|---|---|---|---|---|---|---|
| *Socio-demographic characteristics* | | | | | | | | |
| **Age (years)** | | | | | | | | |
| ≤65 | 386 (49.4) | 254 (45.2) | 12 (44.4) | 31 (55.4) | 30 (53.6) | 37 (78.7) | 11 (73.3) | <0.0001[a] |
| >65 | 396 (50.6) | 308 (54.8) | 15 (55.6) | 25 (44.6) | 26 (46.4) | 10 (21.3) | 4 (26.7) | |
| **Sex** | | | | | | | | |
| Male | 458 (58.6) | 316 (56.2) | 17 (63) | 33 (58.9) | 37 (66.1) | 28 (59.6) | 11 (73.3) | 0.16 |
| Female | 324 (41.4) | 246 (43.8) | 10 (37) | 23 (41.1) | 19 (33.9) | 19 (40.4) | 4 (26.7) | |
| **Health insurance** | | | | | | | | |
| Yes | 687 (88.5) | 527 (94.8) | 24 (88.9) | 47 (83.9) | 48 (85.7) | 22 (46.8) | 8 (53.3) | <0.0001[b] |
| No | 89 (11.5) | 29 (5.2) | 3 (11.1) | 9 (16.1) | 8 (14.3) | 25 (53.2) | 7 (46.7) | |
| **Region** | | | | | | | | |
| Brussels poverty area | 465 (60.2) | 299 (53.5) | 19 (76) | 37 (67.3) | 51 (92.7) | 31 (68.9) | 13 (100) | <0.0001[c] |
| Brussels other | 120 (15.5) | 101 (18.1) | 5 (20) | 7 (12.7) | 2 (3.6) | 3 (6.7) | 0 | |
| Wallonia | 77 (10) | 69 (12.3) | 0 | 3 (5.5) | 0 | 3 (6.7) | 0 | |
| Flanders | 110 (14.3) | 90 (16.1) | 1 (4) | 8 (14.6) | 2 (3.6) | 8 (17.8) | 0 | |
| **Origins of patients** | | | | | | | | |
| Home | 468 (62.2) | 319 (58.4) | 13 (50) | 34 (66.7) | 47 (87) | 35 (76.1) | 9 (75) | <0.0001[d] |
| Nursing home | 112 (14.9) | 90 (16.8) | 9 (34.6) | 7 (13.7) | 2 (3.7) | 1 (2.2) | 0 | |
| Shelters | 20 (2.7) | 4 (0.7) | 2 (7.7) | 1 (1.9) | 1 (1.8) | 7 (15.2) | 2 (16.7) | |
| Other hospital unit | 72 (9.5) | 61 (11.2) | 2 (7.7) | 4 (7.8) | 1 (1.8) | 3 (6.5) | 1 (8.3) | |
| Other hospital | 81 (10.8) | 72 (13.2) | 0 | 5 (9.8) | 3 (5.6) | 0 | 0 | |
| *Pre-existing conditions* | | | | | | | | |
| **Obesity** | | | | | | | | |
| No | 522 (68.4) | 377 (69.3) | 17 (63) | 39 (69.6) | 38 (69.1) | 27 (57.5) | 11 (73.3) | 0.76 |
| Yes | 241 (31.6) | 167 (30.7) | 10 (37) | 17 (30.4) | 17 (30.9) | 20 (42.6) | 4 (26.7) | |
| **Diabetes** | | | | | | | | |
| No | 558 (71.9) | 401 (72.7) | 21 (77.8) | 41 (73.2) | 34 (60.7) | 34 (72.3) | 10 (66.7) | 0.63 |
| Yes | 218 (28.1) | 152 (27.3) | 6 (22.2) | 15 (26.8) | 22 (39.3) | 13 (27.7) | 5 (33.3) | |
| **Hypertension** | | | | | | | | |
| No | 352 (45.3) | 231 (41.5) | 14 (51.8) | 25 (44.6) | 35 (62.5) | 26 (55.3) | 9 (60) | 0.01[e] |
| Yes | 425 (54.7) | 326 (58.5) | 13 (48.2) | 31 (55.4) | 21 (37.5) | 21 (44.7) | 6 (40) | |
| **Cardiovascular disease** | | | | | | | | |
| No | 532 (68.7) | 361 (64.8) | 19 (70.4) | 40 (72.7) | 44 (78.6) | 38 (80.9) | 13 (92.9) | 0.007[f] |
| yes | 243 (31.3) | 196 (35.1) | 8 (29.6) | 15 (27.3) | 12 (21.4) | 9 (19.1) | 1 (7.1) | |
| **Pulmonary disease** | | | | | | | | |
| No | 623 (80.1) | 433 (77.6) | 19 (70.4) | 50 (89.3) | 47 (83.9) | 43 (91.5) | 14 (93.3) | 0.03[g] |
| Yes | 155 (19.9) | 125 (22.4) | 8 (29.6) | 6 (10.7) | 9 (16.1) | 4 (8.5) | 1 (6.7) | |
| **Neoplasia** | | | | | | | | |
| No | 657 (85.3) | 470 (85.3) | 22 (81.5) | 45 (80.4) | 48 (87.3) | 41 (87.2) | 14 (93.3) | 0.84 |
| Yes | 113 (14.7) | 81 (14.7) | 5 (18.5) | 11 (19.6) | 7 (12.7) | 6 (12.8) | 1 (6.7) | |
| *Clinical features* | | | | | | | | |
| **Length of stay (days)** | | | | | | | | |
| Median (EIQ) | 7 (4–15) | 7 (4–15) | 9 (6–16) | 7 (4–15) | 8 (5–15) | 9 (6–14) | 7 (4–16) | 0.30 |
| **ICU** | | | | | | | | |
| No | 638 (82) | 455 (81.4) | 23 (85.2) | 48 (87.3) | 50 (89.3) | 34 (72.3) | 13 (86.7) | 0.35 |
| Yes | 140 (18) | 104 (18.6) | 4 (14.8) | 7 (12.7) | 6 (10.7) | 13 (27.7) | 2 (13.3) | |

*(Continued)*

**Table 1.** (Continued)

| n (%) | Total N = 782 | Belgium n = 562 | EU15 n = 27 | EU28 (without EU15) n = 56 | North Africa N = 56 | Sub-Sahara Africa N = 47 | Middle east N = 15 | p-value |
|---|---|---|---|---|---|---|---|---|
| **Death** | | | | | | | | |
| No | 629 (80.4) | 442 (78.7) | 21 (77.8) | 50 (89.3) | 43 (76.8) | 42 (89.4) | 15 (100) | 0.10 |
| Yes | 153 (19.6) | 120 (21.3) | 6 (22.2) | 6 (10.7) | 13 (23.2) | 5 (10.6) | 0 | |

[a] p<0.0001 Belgium vs sub-Sahara Africa, p = 0.03 Belgium vs middle east;

[b] p = 0.001 Belgium vs EU28, p = 0.006 Belgium vs North Africa, p<0.0001 Belgium vs sub-Sahara Africa and Belgium vs middle east;

[c] p<0.0001 Belgium vs North Africa, p = 0.0003 Belgium vs middle east;

[d] p<0.0001 Belgium vs North Africa, p = 0.02 Belgium vs sub-Sahara Africa;

[e] p = 0.002 Belgium vs North Africa;

[f] p = 0.025 Belgium vs sub-Sahara Africa, p = 0.04 Belgium vs North Africa, p = 0.02 Belgium vs middle east;

[g] p = 0.025 Belgium vs sub-Sahara Africa

Patients from the Middle East were also younger (73.3% under 65 years) and presented a high rate of no health insurance (46.7%) compared to other nationality group. Patients from North Africa had a higher prevalence of diabetes (41.1%) and came mainly from the Brussels deprivation area (92.7%), like Middle Eastern patients (100%). Origin prior admission of North African, Sub–Saharan African, and Middle Eastern patients was mostly from home (87%, 76,1% and 75% respectively). By contrast, a high percentage of EU15 patients came from nursing homes (34.6%), compared to other nationality groups. A significantly higher proportion of patients from sub Saharan Africa and middle east, compared to Belgium, originated from shelters (p<0.0001 and p = 0.006 respectively). Belgians-nationality group presented a higher prevalence of hypertension (58.5%) and cardiovascular diseases (35.1%); EU15 patients also had a higher prevalence of pulmonary disease (29.6%) compared to other nationality groups.

We did not observe any significant differences of ICU admission and mortality rates among the different migrant groups. However, Sub-Saharan African patients had the highest proportion of ICU admission (27.7%) but the lowest mortality rates, with EU28 (10.7%), compared to other nationalities. In contrast North African patients had the highest mortality rates (23.2%) but the lowest proportion of ICU admission (10.7%) among the different nationality groups.

## Risk factors of admission in Intensive Care Units among COVID-19 hospitalized patients

Table 2 shows the univariate and multivariable analysis results, for the association between ICU admission and our social and medical data.

Sex, health insurance, and obesity were independently associated with ICU admission. Male patients who were obese and lacked health insurance had a significantly higher risk of ICU admission. Patients under 65 years had a higher risk of ICU admission compared to older patients (p = 0.01), but after adjusting the excess, the risk decreased significantly. Sub-Saharan African patients had the highest proportion of ICU admission (27.7%) compared to other nationality groups. After adjustment, North African patients had a lower risk of ICU admission compared to Belgian patients: OR(IC95%) 0.3 (0.1–0.8). Neither hypertension, diabetes, neoplasia, cardiovascular and pulmonary diseases were significantly associated with ICU admission. We observed a significantly lower risk of ICU admission for patients from nursing homes compared to those who arrived from home.

**Table 2. Crude rates and odds ratios of the association between ICU admission and risk factors of hospitalized COVID-19 patients.**

| | | | | Cases = 130 / n = 722 | |
| --- | --- | --- | --- | --- | --- |
| | N (%) | OR (CI 95%) | p-value | aOR (CI 95%) | p-value |
| **Sex** | | | | | |
| Male | 96 (21) | 1.6 (1.1–2.4) | 0.01 | 1.7 (1.1–2.5) | 0.02 |
| Female | 45 (13.9) | 1 | | 1 | |
| **Age (years)** | | | | | |
| < 65 | 83 (21.5) | 1 | 0.01 | 1 | 0.25 |
| ≥ 65 | 58 (14.7) | 0.6 (0.4–0.9) | | 0.8 (0.5–1.2) | |
| **Health insurance** | | | | | |
| Yes | 117 (17.1) | 1 | 0.03 | 1 | 0.035 |
| No | 24 (26.7) | 1.8 (1.1–2.9) | | 1.9 (1.0–3.6) | |
| **Origins of patients** | | | | | |
| Home | 78 (16.7) | 1 | <0.0001 | | |
| Nursing home | 6 (5.4) | 0.3 (0.1–0.7) | | | |
| Shelter | 1(5) | 0.3 (0.03–2.0) | | | |
| Other Hospital unit | 26 (36.1) | 2.8 (1.6–4.8) | | | |
| Other Hospital | 24 (30) | 2.1 (1.2–3.7) | | | |
| **Nationality** | | | | | |
| Belgium | 104 (18.6) | 1 | 0.36 | 1 | 0.15 |
| EU15 | 4 (14.8) | 0.8 (0.3–2.2) | | 0.8 (0.2–2.3) | |
| EU28 (without EU15) | 7 (12.7) | 0.6 (0.3–1.45) | | 0.4 (0.1–1.0) | |
| North Africa | 6 (10.7) | 0.5 (0.2–1.25) | | 0.3 (0.1–0.8) | |
| Sub Sahara Africa | 13 (27.7) | 1.7 (0.85–3.3) | | 1.0 (0.4–2.1) | |
| Middle east | 2 (12.5) | 0.6 (0.1–2.8) | | 0.5 (0.1–2.7) | |
| Other | 4 (21.1) | 1.2 (0.4–3.6) | | 0.6 (0.2–2.1) | |
| **Region** | | | | | |
| Brussels poverty area | 80 (17.2) | 0.8 (0.5–1.3) | 0.16 | 0.8 (0.5–1.3) | 0.16 |
| Brussels | 25 (20.8) | 1 | | 1 | |
| Wallonia | 19 (25) | 1.3 (0.6–2.5) | | 1.1 (0.5–2.3) | |
| Flanders | 14 (13) | 0.6 (0.3–1.15) | | 0.5 (0.2–1.0) | |
| **Hypertension** | | | | | |
| No | 63 (17.9) | 1 | 0.91 | 1 | 0.47 |
| Yes | 77 (18.2) | 1.02 (0.7–1.5) | | 1.2 (0.7–1.8) | |
| **Diabetes** | | | | | |
| No | 98 (17.7) | 1 | 0.66 | 1 | |
| Yes | 42 (19) | 1.1 (0.7–1.6) | | 1.2 (0.7–1.9) | 0.44 |
| **Cardiovascular disease** | | | | | |
| No | 105 (19.8) | 1 | 0.07 | 1 | 0.25 |
| Yes | 35 (14.3) | 0.7 (0.45–1.0) | | 0.7 (0.4–1.2) | |
| **Pulmonary disease** | | | | | |
| No | 114 (18.3) | 1 | 0.80 | 1 | 0.73 |
| Yes | 27 (17.4) | 0.9 (0.6–1.5) | | 0.9 (0.5–1.5) | |
| **Obesity** | | | | | |
| No | 84 (16.1) | 1 | 0.04 | 1 | 0.04 |
| Yes | 54 (22.2) | 1.5 (1.0–2.2) | | 1.5 (1.02–2.3) | |
| **Neoplasia** | | | | 0.25 | | 0.40 |
| No | 124 (18.8) | 1 | | 1 | |

(Continued)

**Table 2.** (Continued)

| | N (%) | OR (CI 95%) | p-value | Cases = 130 / n = 722 | |
| --- | --- | --- | --- | --- | --- |
| | | | | aOR (CI 95%) | p-value |
| Yes | 16 (14.3) | 0.7 (0.4–1.3) | | 0.8 (0.4–1.4) | |

*OR* odds ratio, *aOR* adjusted odds ratio, *95% CI* 95% confidence interval

## Mortality risk factors among COVID-19 hospitalized patients

Table 3 shows the univariate and multivariable analysis results, for the association between mortality and our social and medical data. The risk factors are quite different from those linked to ICU admission.

Sex, age, hypertension, cardiovascular disease and neoplasia were significantly associated with mortality. But after adjustment, we observed that only age, sex and hypertension were significantly associated with mortality. Patients older than 65 years were particularly at risk compared to patients under 65 years (OR (IC95%) = 3.7(2.2–6.1), p<0.0001). North African patients had the highest proportion of mortality rate (23.2%), a rate which was close to that of Belgian and EU15 patients (21.4% and 22.2% respectively). By contrast, Sub Saharan African patients had the lowest mortality rate (10,6%). Neither obesity, diabetes, neoplasia, cardiovascular and pulmonary diseases were significantly associated with mortality after adjustment for the other variables. We observed a significant higher risk of mortality in patients coming from nursing homes, compared to those who arrived from home.

## Discussion

This study is the first to analyze a combination of clinical and social data among all patients who were hospitalized for COVID-19during the first wave of the pandemic in poorest area of Brussels-Capital Region, including non-citizens, migrants and refugees. Our research has two main findings: 1/ the social and clinical profile of the patients differs between the nationality groups 2/ some risk factors for ICU admission and mortality were linked to more social conditions as the availability of health insurance and co-morbidities as obesity and hypertension, with a high incidence among people in disadvantaged socio-economic situations.

Older age, male sex and co-morbidities have already been described as risks factor for severe disease and death in patients with COVID-19 [32]. In our study, the lower risk of ICU admission for older patients could be explained by the fact that during the first wave, older patients did not transit through ICUs and do not resuscitate orders. We hypothesized a similar explanation for patients from nursing homes in Brussels. Transfers from nursing homes to hospital were not systematic and suffered delays [33]. Patients arrived at hospital in a critical state and did not transit to ICU. This could explain why we observed a significantly lower risk, but a significantly higher risk of mortality for patients from nursing homes, compared to those who arrived from home.

In our study, obesity and hypertension were significantly associated with ICU admission and mortality, respectively. Obesity was independently associated with ICU but not to mortality. Similar results have been shown in other studies. A French cohort study showed that obesity was a factor in SARS-CoV-2 disease severity, having the greatest impact on patients with a Body Mass Index (BMI) $\geq$ 35 [34]. In New York City, a retrospective analysis has shown that obese patients aged < 60 years were twice as likely to be admitted to acute and critical care, compared to non-obese patients [35]. One of the interesting finding is that we found

**Table 3. Crude rates and odds ratios of the association between mortality and risk factors of hospitalized COVID-19 patients.**

| | | | | Cases = 136 / n = 714 | |
|---|---|---|---|---|---|
| | N (%) | OR (CI 95%) | *p-value* | aOR (CI 95%) | *p-value* |
| **Sex** | | | | | |
| Male | 102 (22.2) | 1.5 (1.1–2.2) | *0.02* | 1.6 (1.0–2.4) | *0.03* |
| Female | 51 (15.6) | 1 | | 1 | |
| **Age (years)** | | | | | |
| < 65 | 32 (8.2) | 1 | *<0.0001* | 1 | *<0.0001* |
| ≥ 65 | 121 (30.5) | 4.9 (3.2–7.4) | | 3.7 (2.2–6.1) | |
| **Health insurance** | | | | | |
| Yes | 140 (20.3) | 1 | *0.07* | 1 | *0.82* |
| No | 11 (12.2) | 0.5 (0.3–1.05) | | 0.9 (0.4–2.0) | |
| **Origins of patients** | | | | | |
| Home | 58 (12.3) | 1 | *<0.0001* | | |
| Care home | 41 (36.6) | 4.1 (2.6–6.6) | | | |
| Shelter | 0 | *nd* | | | |
| Other Hospital unit | 26 (35.6) | 3.9 (2.3–6.8) | | | |
| Other Hospital | 21 (25.9) | 2.5 (1.4–4.4) | | | |
| **Nationality** | | | | | |
| Belgium | 120 (21.4) | 1 | *0.25* | 1 | *0.57* |
| EU15 | 6 (22.2) | 1.05 (0.4–2.7) | | 1.0 (0.3–2.8) | |
| EU28 (without EU15) | 6 (10.7) | 0.4 (0.2–1.05) | | 0.4 (0.2–1.1) | |
| North Africa | 13 (23.2) | 1.1 (0.6–2.1) | | 1.4 (0.3–2.9) | |
| Sub Sahara Africa | 5 (10.6) | 0.4 (0.2–1.1) | | 0.9 (0.3–2.7) | |
| Middle east | 0 | *nd* | | *nd* | |
| Other | 3 (15.8) | 0.7 (0.2–2.5) | | 0.95 (0.25–3.6) | |
| **Region** | | | | | |
| Brussels poverty area | 95 (20.4) | 1.15 (0.7–1.9) | *0.79* | 1.2 (0.7–2.1) | *0.86* |
| Brussels | 22 (18.2) | 1 | | 1 | |
| Wallonia | 15 (19.5) | 1.1 (0.5–2.3) | | 1.0 (0.5–2.4) | |
| Flanders | 18 (16.4) | 0.9 (0.4–1.7) | | 0.9 (0.4–2.0) | |
| **Hypertension** | | | | | |
| No | 44 (12.5) | 1 | *<0.0001* | 1 | *0.01* |
| Yes | 108 (25.4) | 2.4 (1.6–3.5) | | 1.8 (1.1–2.9) | |
| **Diabetes** | | | | | |
| No | 100 (18) | 1 | *0.12* | 1 | *0.70* |
| Yes | 51 (22.9) | 1.3 (0.9–2.0) | | 0.9 (0.6–1.4) | |
| **Cardiovascular disease** | | | | | |
| No | 75 (14) | 1 | *<0.0001* | 1 | *0.16* |
| Yes | 76 (31.1) | 2.8 (1.9–4.0) | | 1.4 (0.9–2.1) | |
| **Pulmonary disease** | | | | | |
| No | 114 (18.2) | 1 | *0.11* | 1 | *0.93* |
| Yes | 37 (23.9) | 1.4 (0.9–2.1) | | 1.0 (0.6–1.6) | |
| **Obesity** | | | | | |
| No | 104 (19.9) | 1 | *0.48* | 1 | *0.79* |
| Yes | 43 (17.7) | 0.9 (0.6–1.3) | | 1.05 (0.7–1.7) | |
| **Neoplasia** | | | | *0.001* | | *0.13* |
| No | 114 (17.2) | 1 | | 1 | |

(*Continued*)

**Table 3.** (Continued)

| | N (%) | OR (CI 95%) | p-value | Cases = 136 / n = 714 | |
| --- | --- | --- | --- | --- | --- |
| | | | | aOR (CI 95%) | p-value |
| Yes | 35 (31) | 2.2 (1.4–3.4) | | 1.5 (0.9–2.5) | |

*OR* odds ratio, *aOR* adjusted odds ratio, *95% CI* 95% confidence interval

hypertension to be associated with COVID-19 mortality, as previously reported [36] but not the other co-morbidities.

Obesity and hypertension are co-morbidities with a high incidence among people in disadvantaged socio-economic situations [37]. In New York City, neighborhoods with higher poverty rates experience disproportionate diabetes and hypertension co-morbidities, and higher rates of COVID-19 infections [14]. Other US studies have shown that co-morbidities such as obesity and cardiovascular diseases, where Blacks and Others are disproportionately over-represented, were a risk factor for COVID-19 related mortality [38, 39]. A number of studies, especially in the US and UK, have also shown that ethnic groups are also over-represented in poor inner-city areas.

In our study, Sub Saharan Africa is the nationality group with the highest risk of ICU admission but with the lowest mortality rate. This group has the highest rate of obesity, which is a risk factor for severe disease, but this is also the youngest nationality group, which could have decreased the mortality risk. The reviewed studies showed that COVID-19 significantly impacted black people across all the outcomes measured, compared to white people [15]. In Belgium, a population based study showed that among men of sub-Saharan African origin aged 40 to 65, mortality increased by 70% compared to the previous year (23).

Poverty and socio-economic status can also influence access to critical care. In our study, patients deprived of health insurance had a higher risk of ICU admission compared to those who had insurance. Among those, patients from sub-Saharan Africa and the Middle East were over-represented (50% without health insurance). Not having health insurance could have contributed to delays in access to tests and healthcare because of the phenomenon of non-take up of health and social security benefits [40]. Our study includes a significant proportion of patients from foreign nationalities who lack health insurance and reside in shelters. Implementing universal health coverage for the entire population, including non-citizens, migrants, and refugees, is an important step in reducing barriers to accessing healthcare for the most vulnerable individuals [41]. Differences observed in COVID-19 hospitalization and mortality rates reflect general trends in racial/ethnic health disparities, which arise from the complex interactions of poverty, access to healthcare and individual factors, such as chronic disease, obesity and hypertension. The area's poverty level was independently associated with a greater risk of hypertension, diabetes and obesity [37]. In addition, crowed housing conditions, urban density, and occupational exposure increased the risk of COVID-19 infections and adverse outcomes [42, 43]. In New York City, 24% of frontline workers live at or below twice the poverty line [44] and Blacks and Hispanics working in key services experienced higher COVID-19 mortality rates than Whites [14, 21].

Our work has a number of limitations. Social data was limited. In hospital records, socio-economic data is not routinely collected, which is a general problem in hospital databases [45]. Moreover, like all hospitals, those of this study are facing a local selectivity of patients that prevents a generalization on the whole Belgian territory. However, the findings of this preliminary research can be useful as a hypothesis for larger research requiring the linkage of hospital

records and other social security records collected by several Belgian statistical institutions. This process in Belgium is complicated: requests approvals and data delivery were delayed, which did not allow us to have information on social and clinical data at the time of the COVID-19 epidemic. During the first wave, only countries such as the US or the UK, which routinely link clinical and social databases, were able to analyze those factors. In future research, a linkage between administrative databases (hospital and social security database) will enable to study in greater detail the social determinants of health associated with COVID-19, including factors such as housing, occupation, education, work conditions, and migrant background. However, our study holds significant value as it includes the analysis of all hospitalized individuals, including non-resident, refugees, migrant, homeless, This population, referred to as "invisible," is not captured in administrative databases and official statistics. The lack of data makes it difficult to conduct studies on this population.

We were also limited in terms of migration background data (we only have the nationality at hospital admission). Additional information on country of birth and parents nationality could have provided valuable insights.

Our statistical analysis was also limited by the numbers in our nationality groups. In order to compare hospitalized patients' socioeconomic profiles, this study must be completed with a second wave study. Epidemic management and control were not the same as during the first wave.

## Conclusion

Our study of patients hospitalized for COVID-19 in two hospitals located in Brussels' poorest area shows that the social and clinical profile of the patients differs between the nationality groups. Moreover, risk factors of ICU admission and mortality are well known risk factors such as sex and age, but also other risks related to social conditions, such as health insurance, obesity and hypertension. The importance of social health determinants has been widely described. But the COVID-19 epidemic has revealed and increased social health inequalities, making it crucially important to routinely collect both clinical and social data.

## Supporting information

**S1 Table. Laboratory parameters of the population hospitalized for COVID-19 by nationality groups.**
(DOCX)

## Acknowledgments

We thank the Fonds national de la Recherche Scientifique (FNRS) and Emmanuel Riviere for editing the manuscript. N.D. is a post-doctorate clinical master specialist of the F.R.S-FNRS.

## Author Contributions

**Conceptualization:** Judith Racape.

**Data curation:** Judith Racape, Anne-Cecile Noel, Jerome Lurel.

**Formal analysis:** Judith Racape.

**Methodology:** Judith Racape, Jean-Christophe Goffard.

**Resources:** Jean-Christophe Goffard.

**Supervision:** Andrea Rea.

**Validation:** Nicolas Dauby, Yves Coppieters, Jean-Christophe Goffard, Andrea Rea.

**Writing – original draft:** Judith Racape.

**Writing – review & editing:** Judith Racape, Nicolas Dauby.

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
