## [Decision Letter · Decision Letter 0]

20 Apr 2023

PGPH-D-23-00305

Social and clinical risk factors associated to hospitalized COVID-19 patients in Brussels’s deprived and multiethnic areas

Dear Dr. Racape,

Thank you for submitting your manuscript to PLOS Global Public Health. After careful consideration, we feel that it has merit but does not fully meet PLOS Global Public Health’s publication criteria as it currently stands. Therefore, we invite you to submit a revised version of the manuscript that addresses the points raised during the review process.

Please respond to each reviewer query - if you do not want to make any edits just explain your reasoning.

We look forward to receiving your revised manuscript.

Kind regards,

Abram L. Wagner, PhD, MPH

Academic Editor

Journal Requirements:

2. In the online submission form, you indicated that "Data will be available upon request from the corresponding author". All PLOS journals now require all data underlying the findings described in their manuscript to be freely available to other researchers, either 1. In a public repository, 2. Within the manuscript itself, or 3. Uploaded as supplementary information.

Additional Editor Comments (if provided):

Reviewers' comments:

Reviewer's Responses to Questions

**Comments to the Author**

1. Does this manuscript meet PLOS Global Public Health’s publication criteria? Is the manuscript technically sound, and do the data support the conclusions? The manuscript must describe methodologically and ethically rigorous research with conclusions that are appropriately drawn based on the data presented.

Reviewer #1: No

Reviewer #2: Yes

Reviewer #3: Yes

Reviewer #4: Yes

Reviewer #5: Yes

2. Has the statistical analysis been performed appropriately and rigorously?

Reviewer #1: No

Reviewer #2: Yes

Reviewer #3: Yes

Reviewer #4: Yes

Reviewer #5: Yes

3. Have the authors made all data underlying the findings in their manuscript fully available (please refer to the Data Availability Statement at the start of the manuscript PDF file)?

Reviewer #1: Yes

Reviewer #2: Yes

Reviewer #3: Yes

Reviewer #4: No

Reviewer #5: Yes

4. Is the manuscript presented in an intelligible fashion and written in standard English?

Reviewer #1: Yes

Reviewer #2: Yes

Reviewer #3: Yes

Reviewer #4: Yes

Reviewer #5: Yes

5. Review Comments to the Author

Reviewer #1: This topic deals with social determinant of health. However the methodology do not to enable reproducibility, which is a key component in research to enable reproducibility. The magnitude of the problem is not reflecting in the introduction section such as incidence rate of Covid in Belgium, mortality rate from Covid. The research design and study eligibility criteria for participants is missing. The result section is not explained in the detailed as the variables are not spelt out.

Author should include a flow diagram of study participant criteria, The figures are very much needed to improve readability and comprehension of detail. The demographic characteristics should be seperated from the hospitalization characteristics

All result should be interpreted.

Reviewer #2: The hyothesis is good enough but the scope of the data is limited. The data adds onto global data on Covid-19 but does does not propose any new information regarding the compounding and risk factors for Covid-19, that is not available elsewhere for similar populations and could otherwise be extrapolated.

Reviewer #3: Thank you for submitting your manuscript to PLOS Global Public Health. Overall, I found the manuscript to be well-written, well-organized, and methodologically sound. The study addresses an important and timely topic, and the results interest many readers. Below are some specific comments and suggestions for improvement:

Introduction:

• The introduction provides an adequate background and rationale for the study. However, it could be strengthened by including a more detailed discussion of the existing literature on COVID-19 and social determinants of health, particularly in the Belgian context.

Methods:

• The methods section is clear and well-organized. However, including more information about the sampling strategy and how participants were recruited for the study would be helpful. For example, was it a case of enrolling all patients found for the period excluding pregnant women? This example may be the actual case, but not stating it may leave the reader perplexed about what took place. Additionally, it would be helpful to know how missing data were handled in the analysis.

Results:

• The results section is well-written and presents the findings.

• The authors could also provide more information about the statistical tests used to assess differences between groups and the significance levels of those tests.

Discussion:

• The discussion section is well-written and interprets the results well.

• The authors should also address the limitations of their study more explicitly, although we recognize the limitations presented by word count for certain article types. If possible, they could also suggest directions for future research in this area in greater detail. Once again, word count may limit the ability to address this.

Conclusion:

• The conclusion is clear and concise.

General comments:

• Overall, the manuscript is well-written and well-organized.

Reviewer #4: Thank you for an interesting deep dive in to the social and clinical risk factors associated with COVID -19 hospitalization in select parts of Brussels. This is needed to drive the discussion on the possible inequities that might drive health and health access in similar pandemics. The manuscript is well written, and clear. It takes a bold and clear look at the possibilities of poorer outcomes for certain nationalities, and highlights deficits in health care outcomes by both clinical and most interestingly, limited social characteristics.

Please find below thoughts and comments that can be addressed to strengthen the already good manuscript.

1. Consider rewriting the title from "Social and clinical risk factors associated to hospitalized COVID-19 patients in Brussels’s deprived and multiethnic areas" to "Social and clinical risk factors associated *with* hospitalized COVID-19 patients in Brussels’s deprived and multiethnic areas." "With" is more appropriate grammatically for clarity than "to"

2. Stylistic and typographical points to aid clarity

A. Please list the specific objectives rather than placing them in a single long sentence. e.g.

1. .....

2. .....

B. Kindly define EU15, EU28 for the international, uninitiated readership. Is the separation of these 2 EU groups of citizens be justified for this study?

C. Line 146- "tata" should read "data"

3. It is very important for the authors to clarify whether they grouped individuals by :ethnicity" or "nationality" which are distinct. Were these nationalities of birth or were, for instance, dark-skinned Belgians of previous African extraction classified as sub-Saharan African in the original dataset? This has huge implications for the study either way.

4. In the authors' experience, do patients with temporary health insurance also experience delays in access to tests and health care equivalent to those without health insurance at all? This will be critical to justifying the inclusion of patients with temporary health insurance in the same category of those with no insurance. Kindly justify this in the methods.

5. A brief background of the hospitals and their location would be helpful to readers. Where exactly are they located, how many beds? It would appear (but is unclear from the body of the text) that these hospitals are located in vulnerable settings and are pretty accessible to immigrant communities. What is the profile/diversity of their patients at baseline? Just a brief introduction to these 2 facilities would be helpful.

6. Reading the abstract, it is very easy to assume that mortalities are higher for patients from sub-Saharan Africa, even though this is not the case in the body of the manuscript. It is worth mentioning in the abstract around lines 47-48 "Patients from Sub-Saharan Africa were younger, had a higher prevalence of obesity, lacked health insurance, and had the highest proportion of Intensive Care Unit (ICU) admission (27.7%), *but the lowest mortality rates, compared* to other nationality groups." See line 191/192

Thank you for this thoughtful manuscript.

Reviewer #5: This paper is well written. However, the authors could consider relating the study to the sustainable development goals (SDG) 3.8, which enjoins member countries to implement and promote the universal health coverage, including financial risk protection, access to quality essential health-care services and access to safe, effective, quality and affordable essential medicines.

6. PLOS authors have the option to publish the peer review history of their article (what does this mean?). If published, this will include your full peer review and any attached files.

**Do you want your identity to be public for this peer review?** For information about this choice, including consent withdrawal, please see our Privacy Policy.

Reviewer #1: No

Reviewer #2: No

Reviewer #3: **Yes: **Alain Casseus

Reviewer #4: No

Reviewer #5: **Yes: **BOTHA, Nkosi Nkosi

---

## [Decision Letter · Decision Letter 1]

9 Jun 2023

Social and clinical risk factors associated with hospitalized COVID-19 patients in Brussels’s deprived and multiethnic areas

PGPH-D-23-00305R1

Dear Mrs Racape,

We are pleased to inform you that your manuscript 'Social and clinical risk factors associated with hospitalized COVID-19 patients in Brussels’s deprived and multiethnic areas' has been provisionally accepted for publication in PLOS Global Public Health.

Best regards,

Julia Robinson

Executive Editor

Reviewer Comments (if any, and for reference):

Reviewer's Responses to Questions

**Comments to the Author**

1. If the authors have adequately addressed your comments raised in a previous round of review and you feel that this manuscript is now acceptable for publication, you may indicate that here to bypass the “Comments to the Author” section, enter your conflict of interest statement in the “Confidential to Editor” section, and submit your "Accept" recommendation.

Reviewer #4: All comments have been addressed

Reviewer #5: All comments have been addressed

2. Does this manuscript meet PLOS Global Public Health’s publication criteria? Is the manuscript technically sound, and do the data support the conclusions? The manuscript must describe methodologically and ethically rigorous research with conclusions that are appropriately drawn based on the data presented.

Reviewer #4: Yes

Reviewer #5: Yes

3. Has the statistical analysis been performed appropriately and rigorously?

Reviewer #4: Yes

Reviewer #5: Yes

4. Have the authors made all data underlying the findings in their manuscript fully available (please refer to the Data Availability Statement at the start of the manuscript PDF file)?

Reviewer #4: No

Reviewer #5: Yes

5. Is the manuscript presented in an intelligible fashion and written in standard English?

Reviewer #4: Yes

Reviewer #5: Yes

6. Review Comments to the Author

Reviewer #4: All comments have been satisfactorily addressed.

Reviewer #5: All issues raised have been adequately addressed.

7. PLOS authors have the option to publish the peer review history of their article (what does this mean?). If published, this will include your full peer review and any attached files.

**Do you want your identity to be public for this peer review?** For information about this choice, including consent withdrawal, please see our Privacy Policy.

Reviewer #4: No

Reviewer #5: **Yes: **BOTHA, Nkosi Nkosi
